# Road Fatalities in Children Aged 0–17: Epidemiological Data and Forensic Aspects on a Series of Cases in a Single-Centre in Romania

**DOI:** 10.3390/children11091065

**Published:** 2024-08-30

**Authors:** Ştefania Ungureanu, Veronica Ciocan, Camelia-Oana Mureșan, Emanuela Stan, Georgiana-Denisa Gavriliţă, Alexandra Sirmon, Cristian Pop, Alexandra Enache

**Affiliations:** 1Doctoral School, “Victor Babes” University of Medicine and Pharmacy Timisoara, 2 Eftimie Murgu Square, 300041 Timisoara, Romania; stefania.ungureanu@umft.ro (Ş.U.); emanuela.stan@umft.ro (E.S.); denisa.tincu@umft.ro (G.-D.G.); 2Discipline of Forensic Medicine, Bioethics, Deontology and Medical Law, Department of Neuroscience, “Victor Babes” University of Medicine and Pharmacy Timisoara, 2 Eftimie Murgu Square, 300041 Timisoara, Romania; muresan.camelia@umft.ro (C.-O.M.); enache.alexandra@umft.ro (A.E.); 3Timisoara Institute of Legal Medicine, 1A Ciresului Street, 300610 Timisoara, Romania; 4Ethics and Human Identification Research Center, Department of Neuroscience, Discipline of Forensic Medicine, Bioethics, Deontology and Medical Law, “Victor Babes” University of Medicine and Pharmacy Timisoara, 2 Eftimie Murgu Square, 300041 Timisoara, Romania; 5“Pius Branzeu” Emergency County Clinical Hospital Timisoara, 156 Liviu Rebreanu Bld., 300723 Timisoara, Romania; alexandra.sirmon@gmail.com; 6Residency Program in Epidemiology, “Victor Babes” University of Medicine and Pharmacy Timisoara, 2 Eftimie Murgu Square, 300041 Timisoara, Romania; 7Department of Mechatronics, Faculty of Mechanical Engineering, University Politehnica Timisoara, 1 Mihai Viteazu Bld., 300222 Timisoara, Romania; cristian.pop@upt.ro

**Keywords:** road traffic accidents, children, medico-legal autopsy, risk factors, preventive measures

## Abstract

Introduction: Road Traffic Accidents (RTAs) are the leading cause of premature death in young people aged 5–29. Globally, 186,300 children aged 9 years and under die from RTAs each year. Romania had the highest mortality rate in children aged 0 to 14 for 2018–2020. This study aimed to assess the involvement of children aged 0–17 years in fatal RTAs by analyzing medico-legal autopsy records in a 5-year period at Timisoara Institute of Legal Medicine (TILM), Romania. Materials and Methods: A retrospective analysis of medico-legal autopsy records of road fatalities in children aged 0–17 years, from TILM in a 5-year period (2017–2021), was conducted. Results: Of all medico-legal autopsies in the 5-year period, 23 cases (5.8%) involved road fatalities in children aged 17 and under. Preschoolers accounted for 10 cases, followed by the age group 15–17 years (*n* = 9). Most children sustained fatal injuries as passengers (*n* = 13), followed by child pedestrians (*n* = 7). This research follows four representative cases, each being a different type of child road fatality regarding the type of road user, the age of the victim, and the involvement of other risk factors. Conclusions: Our findings emphasize the tragedy of road fatalities in children and the need to determine risk factors and prevention strategies to reduce the enormous global crisis involving these vulnerable victims.

## 1. Introduction

According to the World Health Organization (WHO), as of 2018, approximately 1.35 million individuals are fatally injured each year in Road Traffic Accidents (RTAs), which represent the eighth leading cause of death for people of all ages [1,2]. The WHO report from 2023 showed that 1.3 million people die and millions more are disabled or injured each year as a result of RTAs around the world [3].

Road traffic injuries are the leading cause of premature death in young people aged 5–29 years in the WHO European Region [4]. They also represent the eighth leading cause of death for children aged 4 and under [5]. Motor vehicle accidents (MVAs) represent the most common cause of morbidity and mortality worldwide in adolescents [6].

Globally, 186,300 children aged 9 years and under die from RTAs each year [7].

The main risk factors for RTAs are considered to be as follows: having an urban speed limit higher than 50 km/h, a blood alcohol concentration (BAC) while driving of higher than 0.05 g/dL, and the refusal to use compulsory safety equipment such as seat belts for all car occupants, child restraint systems, and helmets for cyclists or motorized two-wheelers. When addressing the components with a role in RTAs, socioeconomic determinants are also important [5]. Low- and middle-income countries (LMICs) account for up to 95% of road traffic mortalities in children globally, consuming substantial health sector resources [2,7]. Even in high-income countries (HICs) in Europe, children from deprived families are at much greater risk than those of wealthy families. In addition, as stated above, risk factors such as alcohol and speed need to be considered [5].

When addressing the number of road fatalities in children per million inhabitants aged 0 to 14 per country in the European Union (EU) for 2018–2020, Romania had the highest mortality rate for children, alongside Bulgaria and Latvia (Figure 1) [8].

The relative mortality rate, which is calculated by dividing the mortality rate for children by the mortality rate for all ages, puts Romania in third place, with 0.43; the average for EU countries is 0.27. This means that in Romania, children are more likely to be victims of fatal RTAs in comparison to the European average. Additionally, Romania scores above average for the number of road fatalities among children as a proportion of total fatalities [8]. This calls for more action towards understanding the mechanisms and outcomes of traumatic injuries in children for building further effective prevention strategies [9].

Road traffic deaths have been predicted to become the seventh main cause of death worldwide by 2030 if urgent action is not taken [10]. A special attention needs to address vulnerable road users (pedestrians, cyclists, and users of motorized two-wheelers), since they are at greater risk of serious injuries in road crashes because they do not have a protective shell [5,11]. For these victims, traumatic injuries occur mostly at the cranial level, as well as in both upper and lower limbs, which confirms the need for cephalic protection devices. The use of helmets plays a fundamental role in reducing the burden of craniofacial and maxillofacial injuries [12]. Using helmets and other protective equipment such as seat belts and child car protective systems significantly reduces mortality [13]. For the European Region as a whole, vulnerable road users account for 39% of road traffic deaths [5]. Children represent vulnerable road victims, especially in LMICs, where the rate of road fatalities is almost six times higher compared to HICs [14]. According to the European Road Safety Observatory in 2022, children had a very high death toll among the most vulnerable modes of transport: 32% of children killed were pedestrians, while 13% were cyclists. In Romania and Greece, more than one in two fatalities among children occurs when they are a pedestrian [8]. Because of their partial developmental capacity to perceive road and traffic threats, children and youth pedestrians (2–20 years) are at particular risk of a pedestrian motor vehicle collision (PMVC) [15]. In 2016, there were approximately 72,000 pedestrian fatalities among children and youth (0–20 years) worldwide [11]. In America, pedestrian injuries are the third leading cause of injury-related death for boys and girls with ages between 5 and 14 years old [15].

In addition to the lost lives of children, the financial losses from road accidents represent a huge burden for the families of the victims and their countries [16]. These losses emerge from the costs of treatment for the victims and the reduced or lost productivity for those killed or disabled by their injuries, as well as family members who need to take time off work to care for the injured [17]. The global cost of RTAs is approximate to be over half a trillion USD annually [18]. The burden of transport injuries is disproportionally spread worldwide; almost 85% of the global road traffic injury (RTI) deaths and 90% of disability-adjusted life years occurs in LMICs, which have a death rate of 27.5 per 100,000 population, up to three times higher than that in HICs. [14].

Seeing that for 2018–2020, Romania had the highest mortality rate for children in road fatalities [8], an understanding of fatal RTAs that happen in this country and that involve children may bring some light to this phenomenon and unveil the risk factors that lead to such accidents. Thus, analyzing the road fatalities in children in Romania and the circumstances of RTAs may result in developing protective measures aimed at reducing road deaths in children. Since in Romania it is mandatory to perform a medico-legal autopsy in all road deaths, only by analyzing these autopsy records could we understand all the factors that lead to RTAs [19].

Medico-legal autopsies have a crucial role in the epidemiological assessment of RTAs [20]. They help determine the cause and manner of death, time since death, and circumstances of death [21]. In Romania, it is mandatory to perform a medico-legal autopsy in cases of violent or suspicious death or when the cause of death is not known. This relates to all road deaths, disregarding the period of survival from the accident until the victims’ death [19].

This study aimed to assess the involvement of children aged 0–17 years in fatal RTAs by analyzing the medico-legal autopsy records in a 5-year period (2017–2021) at Timisoara Institute of Legal Medicine (TILM) in Timis county, Romania. Regarding road fatalities in children, attention needs to be focused on the assessment of the risk factors that lead to fatal traffic injuries and to the development of preventive measures and interventions targeted at reducing RTAs. The burden of road fatalities in children highlights the need for identification of such risk factors and prevention strategies to reduce the enormous global crisis in these victims.

## 2. Materials and Methods

This study constitutes a retrospective analysis of medico-legal autopsy records from TILM in a 5-year period (2017–2021). The focus of this study is to assess the involvement of children aged 0–17 years old in fatal RTAs with an emphasis on sustained traumatic injuries for identifying the risk factors and associated prevention strategies with specific interventions targeted at reducing RTAs. At TILM, medico-legal autopsies are performed for victims of all violent, suspicious, or unknown-cause deaths that occur in Timis county, Romania. RTAs victims are accounted for as violent deaths; therefore, a medico-legal autopsy is performed.

This study was conducted in accordance with the Declaration of Helsinki and was approved by the Research Ethics Committee of “Victor Babes” University of Medicine and Pharmacy Timisoara registered under the decision number 88/19.12.2022 rev 2024.

Inclusion criteria: medico-legal autopsy records at TILM of children aged between 0 and 17 years old that were involved in RTAs in a 5-year period (2017–2021).

Exclusion criteria: medico-legal autopsy records at TILM of people that were not victims of RTAs, were not children (were not younger than 18 years old), and records not pertaining to the period mentioned in the inclusion criteria above were excluded.

Study tool: The medico-legal autopsy records for children aged between 0 and 17 years old that had died in a RTA were then retrieved and grouped according to the victims’ demographics (age and gender), date of the accident, type of road user (passenger, pedestrian, cyclist, motorcyclist), traumatic injuries sustained in the accident, the period of hospitalization for victims that did not die at the crash site, and blood alcohol concentration (BAC). The subjects were further stratified into five groups based on age intervals: 0–3 years, 4–6 years, 7–10 years, 11–14 years, and 15–17 years. A thorough and meticulous analysis of traumatic injuries was performed regarding the anatomical region involved. The data for fatal RTAs involving children with ages between 0 and 17 years old in a 5-year period were collected and entered into an Excel spreadsheet (Microsoft Office 365 suite) and analyzed using descriptive statistics.

## 3. Results

Between 1 January 2017 and 31 December 2021, a total of 23 medico-legal autopsies involving children aged 17 and under who had died in RTAs were performed. This represents 0.61% of all medico-legal autopsies at TILM in the 5-year period (*n* = 3752). This number also represents 5.8% of all RTA autopsies (*n* = 395).

### 3.1. Yearly Distribution

Table 1 shows the yearly distribution of cases within each age group in the 5-year period. As illustrated, fatally injured children were the least involved among all age groups (*n* = 23).

### 3.2. Summary and General Characteristics of All 23 Road Fatalities in Children Aged 0–17 Years Old

Table 2 illustrates a summary of all road fatalities in children that were autopsied at TILM (*n* = 23), shown in a chronological order regarding the date of the accident. The main items that were studied are: age and gender of the victim, type of road user, circumstances that caused the road accident, whether the victim was hospitalized or not and the period of hospitalization, external and internal traumatic injuries, cause of death, and BAC.

### 3.3. Age and Gender Distribution

The subjects were further stratified into five groups based on age: 0–3 years (*n* = 5), 4–6 years (*n* = 5), 7–10 years (*n* = 2), 11–14 years (*n* = 2), and 15–17 years (*n* = 9) (Table 3). As observed, most cases were in the 15–17 (*n* = 9) age group, with the lowest involvement in the age groups 11–14 and 7–10 years (*n* = 2). Preschoolers (under the age of 7) accounted for 10 cases, distributed equally in the 0–3 and 4–6 years age groups. The ages span from a minimum of 3 months to a maximum of 17 years.

Regarding the gender distribution, the male to female ratio was 1.3:1, with 13 males (56.5%) and 10 females (43.5%).

### 3.4. Type of Road User

When analyzing the RTA victims regarding the type of road user, most children sustained fatal injuries as passengers (*n* = 13). Child PVMCs accounted for seven cases. Among the victims, we also noted two fatally injured children on a motorcycle, one as a driver and the other as a passenger. Between road fatalities, one cyclist was noted.

Table 3 illustrates the distribution of cases among age groups and type of road users.

**Table 3 children-11-01065-t003:** Age group distribution and types of road users of the study population for children (0–17 years old) that were fatally injured in RTAs and had a medico-legal autopsy at Timisoara Institute of Legal Medicine (TILM) in Timis county, Romania, in a 5-year period (2017–2021).

Age Group (Years Old)	0–3	4–6	7–10	11–14	15–17	Total
Number of cases	5	5	2	2	9	23
**Type of road user**	**Passenger**	**Pedestrian**	**Cyclist**	**Motorcyclist**		**Total**
Number of cases	13	7	1	2		23

### 3.5. Cause of Death

The cause of death for most cases (*n* = 15) was represented by polytrauma, with various associations of the following traumatic injuries: injuries to the head (skull fractures, subarachnoid hemorrhage, subdural hematoma, extradural hematoma, cerebral contusions, cerebral lacerations, contusions of the brain stem or cerebellum, intraventricular hemorrhage, intracerebral hematoma), thoracic injuries (hemothorax, rib and sternal fractures, lung contusions, lung lacerations, tears of thoracic blood vessels, tears of the heart muscle and pericardium), abdominal trauma (hemoperitoneum, hepatic lacerations, splenic lacerations, renal lacerations, hepatic contusions, enteral contusions, renal contusions, pancreatic contusions, pancreatic hematoma, mesenteric tears), trauma to the spine (cervical spine fractures, atlanto-occipital dislocation), and fractures of upper and lower limbs and os coxae. The other eight cases presented cranio-cerebral trauma as the cause of death.

### 3.6. Blood Alcohol Concentration

During medico-legal autopsies, blood samples were collected from the victims to determine the BAC. From all the victims (*n* = 23), in 15 cases blood sampling was not performed either because the children were hospitalized for more than 24 h (*n* = 9) or their age was under 6 years old and therefore the police did not require the determination of BAC. However, in eight cases, the concentration of alcohol in blood was determined; seven cases had negative results, but one case presented a BAC of 1.05 g per 1000 milliliters of blood.

### 3.7. Case Presentations

As case presentations, four suggestive cases have been selected for presentation, each representing a different type of road fatality in children regarding the type of road user, the age of the victim, and the involvement of other risk factors such as alcohol consumption, reckless driving, unsupervised children, and the lack of protective equipment. In all cases, according to Romanian law, seeing that the victims’ deaths occurred due to a traumatic event, such as being involved in RTAs (violent deaths), medico-legal autopsies were required by the police. The autopsies established the cause and manner of death for the victims, the sustained external and internal injuries, and the mechanisms of injury. Additionally, the police asked for the determination of BAC in some cases.

We selected these four cases because we consider them highly relevant to this study, i.e., for identifying the risk factors that lead to fatal road accidents and recognizing potential preventive measures and interventions targeted at reducing RTAs. In each of these cases, we can observe a summation of risk factors that were involved in the accidents, such as type of road user, victim’s age in relation to type of road user, reckless driving in spite of the use of protective equipment (3-month-old baby that was involved in a RTA as a back seat passenger who was strapped in a child car seat while the driver was driving recklessly and, while overtaking a car, hit a light pole and then the car was projected into a building); alcohol consumption in child pedestrians (17-year-old pedestrian who was hit by a bus while intoxicated at 4:40 a.m.); victim’s age in relation to type of road user (4-year-old cyclist who was crossing the street and was not seen by the driver while veering right, maybe because of the small stature of children + the child was on a bicycle + the child was not supervised by the parents); and the importance of protective equipment such as helmets while using a motorcycle for all users not just drivers (17 years old passenger on a motorcycle).

#### 3.7.1. Case 1

##### Background

We present the case of a 3-month-old male that was involved in an RTA as a backseat passenger strapped into an infant car seat. The RTA happened at 2:00 p.m. The victim’s car was driven by the father of the child who, while overtaking a car, hit a light pole and was then projected into a building. Following the event, the infant was hospitalized at the Children’s Hospital in Timisoara on the Pediatric Surgery and Orthopedics wing, where he remained for only one day before he succumbed to his injuries. On admission, an emergency orotracheal intubation with mechanical ventilation was performed. The patient presented hemorrhagic and traumatic shock, with cardiac rate of 156 beats per minute and pale skin and mucous membranes. Blood volume resuscitation was initiated. On the examination of lower limbs, a right femur open fracture was noted with ischemia present due to the severed femoral artery, associated with a left femur open fracture and right tibial open fracture. Additionally, severe lacerations were noted on the external genital organs. The patient underwent emergency surgery, performing open reduction of the right femur fracture using an intramedullary osteosynthesis with Kirschner wires, debridement of the lacerations, and the reconstruction of the femoral artery using an autologous saphenous graft. However, the postoperative recovery was unfavorable and the following morning the patient presented with cardiac arrest that did not respond to resuscitation maneuvers.

##### Medico-Legal Autopsy

The next day, a medico-legal autopsy was performed at TILM. The external examination of the victim’s body noted bruises and excoriations located on the head, thorax, abdominal wall, and lower limbs. Moreover, sutured lacerations were noted on the lower limbs and external genital organs, there were postoperative sutured incisions on both lower extremities, and two drain tubes were placed on the right thigh and in the left inguinal region (Figure 2).

The internal examination of the body revealed hemorrhage in the pericranium, cerebral edema, blood in the abdominal cavity, and hemorrhage in the pelvic tissues (Figure 3). The autopsy concluded that the victim’s death was violent, the cause of death being polytrauma resulting in open fractures of the lower extremities.

#### 3.7.2. Case 2

##### Background

We present the case of a 17-year-old male that was found dead on the side of a county road at 4.40 a.m. From the police investigation, it was detailed that he was hit by a bus in an RTA and died at the crash scene. According to the bus driver’s statement, he tried to avoid what had seemed to be the body of a dead animal on the right side of the road, but, by doing this, he was not able to notice the victim on the left side of the road and could not stop the bus in time to avoid the impact.

##### Medico-Legal Autopsy

As part of the police investigation, the police asked for the determination of BAC. The following day, a medico-legal autopsy was performed at TILM. The external examination of the victim’s body revealed an extensive laceration involving the cephalic extremity, the skull, the dura mater, and the brain tissue, with lack of substance and the expulsion of brain tissue with various bone fragments. Excoriations and lacerations on the thoracic anterior wall, abdominal anterior wall, lumbar regions, sacral region, and upper and lower limbs were noted. In addition, an open fracture of the left clavicle, closed fracture of the left humerus, closed double fracture of the right femur, and a closed fracture of the left tibia were also determined.

The internal examination of the body revealed hemorrhage in the pericranium, numerous skull fractures with bone fragments, cerebellar lacerations, pulmonary contusions, blood in the abdominal cavity, hepatic lacerations, and contusions and hemorrhage surrounding the right kidney (Figure 4).

Blood samples were collected for determining the BAC. The analysis identified a BAC of 1.05 g per 1000 milliliters of blood. The medico-legal autopsy concluded that the victim’s death was violent, the cause of death being polytrauma resulting in a complex cranio-cerebral injury (“skull explosion”) with cerebral and cerebellar lacerations, thoracic trauma (pulmonary contusions), abdominal trauma (hepatic lacerations and contusions), and fractures of upper and lower limbs.

#### 3.7.3. Case 3

##### Background

We present the case of a 4-year-old male that was involved in an RTA as a cyclist that was hit by a car. The RTA happened at 7:45 a.m. The collision between the car and the victim happened when the car entered the street while veering to the right and hit the victim, who was crossing the street at a crosswalk. Following the event, the victim was taken to the emergency room by an ambulance. During the transport to the hospital, the patient underwent an emergency orotracheal intubation and was manually ventilated. At the hospital, the child was in a critical condition, having cerebrospinal fluid leakage through the ears and nose. The electrocardiogram noted pulseless electrical activity. The patient was cardiopulmonary resuscitated with chest compressions and administration of medication; however, the patient did not respond to the maneuvers and was declared dead.

##### Medico-Legal Autopsy

The medico-legal autopsy was performed the next day. The external examination of the victim’s body noted bruises and excoriations located on the face, neck, and upper and lower limbs. Pale skin and ear bleeding were also observed (Figure 5).

The internal examination of the body revealed hemorrhage in the pericranium, skull fractures, subarachnoid hemorrhage, cerebral edema, and cerebral, cerebellar, and intraventricular hemorrhage (Figure 6).

The autopsy concluded that the victim’s death was violent, the cause of death being cranio-cerebral trauma resulting in skull fractures, subarachnoid hemorrhage, cerebral edema, and cerebral, cerebellar, and intraventricular hemorrhage.

#### 3.7.4. Case 4

##### Background

We present the case of a 17-year-old female that was involved in an RTA as a passenger on a motorcycle. The RTA happened at 9:30 p.m. on an urban road. Following the event, the victim was transported to the closest hospital where she underwent an emergency orotracheal intubation, was sedated, and imagistic investigations were performed. Her Glasgow Coma Scale (GCS) score was 3. She was then transferred to the County Hospital in Timisoara. On admission into the intensive care unit, the GCS score was 5, she was orotracheal intubated, mechanically ventilated, and hemodynamically stable. The patient presented bruises around the eyes and on the lower extremities and excoriations on the face, thoracic wall, and upper and lower limbs. The day following admission, the patient underwent neurosurgery, which evacuated a cerebellar subdural hematoma. Despite complex and adequate treatment, due to the severity of the traumatic injuries, the patient died after 36 days of hospitalization.

##### Medico-Legal Autopsy

A medico-legal autopsy was performed at TILM, the following day. The external examination of the victim’s body noted recent scars on the head, thoracic wall, and upper and lower limbs. Additionally, in the occipital region a postoperative scar was observed. The internal examination of the body revealed hemorrhage in the pericranium, the craniectomy orifice, cerebral edema, the remains of subdural hematoma, subarachnoid hemorrhage, and cerebral contusions (Figure 7).

The autopsy concluded that the victim’s death was violent, the cause of death being craniocerebral trauma resulting in subdural hematoma that was evacuated, subarachnoid hemorrhage, and cerebral contusions.

## 4. Discussion

This study assessed the involvement of children in fatal RTAs in the western part of Romania by analyzing medico-legal autopsy records from TILM over a period of five years (2017–2021). Investigating road fatalities in children can offer great and valued information for the development of protective measures and to inhibit the role of risk factors in RTAs.

To the best of our knowledge, this is the first study that analyzes road fatalities in children in the western part of Romania over a 5-year period by studying medico-legal autopsy records. Since Romania had the highest mortality rate for fatally injured children in RTAs (aged 0–14) in European Region countries per million inhabitants in 2018–2020, this study will bring new information regarding the matter and will answer questions related to the circumstances, risk factors, and preventive measures regarding RTAs. Evidence derived from this study will help us to better understand the involvement of children in road fatalities and to develop child road injury prevention initiatives and intervention strategies to reduce child mortality.

By analyzing all 23 cases and the circumstances that led to RTAs, we observed avoidable situations that need to be addressed when thinking of initiatives targeted at lowering road fatalities in children and in the general population. The main risk factors that need to be taken into consideration represent human factors, mainly related to the driver. Driving while intoxicated, speeding, and not adhering to the rules of the road cause most road accidents [5]. Reckless and aggressive driving, such as overtaking cars in an unsafe manner, steering into oncoming traffic, or even driving on the wrong side of the highway, were situations that were responsible for the RTAs presented in this study. Moreover, losing control of the vehicle, maybe because of driving too fast or by not paying enough attention to the road, represents a recipe for disaster. Situations like not being able to regain control of the vehicle and hitting a pedestrian, cyclist, or object on the street (bridge heads, light poles, buildings), were seen in the present study. These situations could have been avoided if the drivers had been driving safely. Another human factor that needs to be considered is the actions of pedestrians. Whether because of their age and their lack of sense of danger or their small stature, children represent ideal victims that may venture into traffic regardless of the consequences [11,14,15]. The most important role is played by the parents, who must supervise children’s activities near the road and not let children roam free. Situations like children being hit by cars that were veering or parking were discovered in the present study. Additionally, children hit by cars while being on the side of the road alone when drivers lost control of their vehicles represent situations in which both the driver and the victims can be held accountable for RTAs.

In the following lines we address the four cases we presented by analyzing the circumstances that led to the road accidents.

The first case we illustrated draws attention to the importance of protective measures and safe driving conduct to protect child victims that are only several months old. Using child car restraints is compulsory in all HICs, but legislation is lacking in 30% of LMICs [5]. Child retention systems, such as safety seats for infants and younger children and booster seats for older children, represent the most important individual-level measure to reduce MVC fatalities among infants and children [22]. The correct implementation and use of these protective systems reduces MVC mortality by 71% in those under 1 year old, by between 54% and 80% in 1- to 4-year-olds, and by 19% in 8- to 12-year-olds. However, the circumstances of the accident reveal that the use of child protective systems is not sufficient. As we can see, other risk factors need to be addressed, such as vehicle safety checks with periodic inspections, speeding, or alcohol consumption. Risky driving conduct, including impairment, distraction, speeding, and aggressive driving, has a role in increasing road injury risk [11]. This example illustrates aggressive driving such as the unsafe overtaking of cars.

The second case we presented was that of an inebriated fatally injured 17-year-old pedestrian who was hit by a bus. This highlights the role alcohol consumption plays in RTAs, even in traffic accidents involving children. The period between childhood and emerging adulthood (adolescence) is characterized by increased sensation seeking and risk-taking behavior such as alcohol and drug use [23]. Studies confirm that alcohol-impaired driving is an established independent risk factor for RTAs; however, alcohol consumption among pedestrians is also considered a risk factor and needs to be addressed more seriously [11,23]. Alcohol use leads to impairment and poor judgement, which raises the odds of an RTA. It also increases reaction times, lowers caution, and reduces visual acuity [24]. As with drivers, a pedestrian’s risk of crash involvement rises with an increasing BAC. For example, 35% of the pedestrians involved in road fatalities in the United States in 2009 had a BAC above 0.08 g/dL, in comparison to 13% of drivers [23,24]. Therefore, a road safety strategy is represented by controlling walking while impaired by alcohol [25]. Speed, alcohol, the absence of infrastructure resources for pedestrians, and the ineffective visibility of pedestrians can be considered the main risk factors for pedestrian fatalities [26]. In Serbia, the combination of cutting back driving under the influence of alcohol with the control of pedestrians’ BAC levels was proposed as a measure to additionally decrease pedestrian fatalities [27].

The third case we presented, a four-year-old cyclist hit by a car, draws attention to unsupervised children on the road and the dangers they are exposed to. Parents need to be extra careful in supervising young children’s outdoor activities, such as bike riding. Other fatal child pedestrian injuries are represented by driveway incidents, mainly involving sports utility or four-wheel-drive-type vehicles; parents with young children should be aware of the dangers that driveways pose [15]. In the presented case, the child was crossing the street at a crosswalk when he was hit by a car while the car was entering the street by veering right. This draws attention to unsupervised children and to the driver’s lack of attention to the road. Distractions for the driver, including talking on a cell phone, texting, or performing other tasks such as eating while driving, were associated with road fatalities in children. Children are at greater risk because of their developing cognitive ability and their small size, which makes them less visible and more prone to fatal injuries in collisions [11]. They often exhibit immature behavior and they do not have a proper awareness of traffic risks [16]. Studies have reported that boys are more likely to run or play near traffic and to choose riskier routes across intersections [16]. Craniocerebral trauma as cause of death emphasizes the vulnerability and low physical tolerance of a child’s body in high-impact collisions [14]. In Romania, a bicycle helmet is mandatory for all cyclists under the age of 16. [28]. Preventive measures like interventions aimed at improving pedestrian and cyclist visibility, as well as area-wide traffic calming has been effective when implemented to avoid RTAs happening [2]. Child pedestrian fatal injuries occur mainly in residential zones, on the same street as or near the child’s home [15]. Enhancement in both infrastructure and automobile safety, for example lowering the speed limit in zones where children and motorized traffic are in close proximity and introducing autonomous emergency braking systems with cyclist/pedestrian recognition will help increase children’s protection from traffic [25].

The fourth case we illustrated highlights the importance of protective equipment usage while riding a motorcycle as a passenger. The motorcycle has proved to be a hazardous mode of transportation, yet there is a rising use of this means of transport because it is cheaper to acquire and easier to meander through heavy traffic in urban regions [29]. Compared to other types of road users per mile travelled, motorcycle users have a 34-fold higher risk of death in an RTA [10]. Wearing a helmet while riding a motorcycle protects the brain and face from serious injury in case of a crash and can save lives. A comprehensive law on helmet use applied to all riders of all motorized two-wheelers on all roads irrespective of age, religion, or engine size would reduce fatalities. However, only half the countries in the European Region have such a law [5]. Road-related head trauma and the escalated risk of sustaining fatal road injuries are mainly associated with the lack of compliance with safety measures such as the use of helmets and child restraining systems [14]. An epidemiologic study conducted by Barzegar et al. showed that head trauma was more prevalent and represented the major cause of death in motorcyclists and that only a small proportion of motorcycle crash victims (37.4%) used helmets, while the majority of victims (62.6%) were not using helmets at the time of crash [10]. Wearing a motorcycle helmet can reduce the risk of death and head trauma by nearly 42% and 69%, respectively [30]. According to the 2023 WHO report on road safety, proper helmet use can lower the risk of mortality in RTAs by more than six times and the risk of brain injury by up to 74% [3]. Therefore, more attention should be focused on the use of safety and protective devices. Education campaigns as well as law enforcement should prioritize the encouragement of motorcycle users to wear helmets, both drivers and their passengers [10].

To summarize, road traffic deaths and injuries constitute a major but neglected public health challenge that requires collaborative efforts for effective and sustainable preventive campaigns [17]. MVC deaths are mostly predictable and avoidable and can be lowered with proper public health involvement [31]. As seen in the cases we presented, the road deaths of these children could have been avoided if drivers respected the rules of the road and had not displayed unsafe and aggressive driving conduct, if unsupervised children were not left alone in the street, if safety equipment was used, and if alcohol consumption was not involved. Therefore, there is a need for targeted and directed intervention approaches to reduce RTIs [32]. Interventions for the prevention of RTAs should be directed towards vulnerable road users but adapted for all other road users [16,30]. These interventions fall into different categories, such as public awareness and education, legislation, and enforcement [2]. Figure 8 illustrates our adaptation of the logic model presented by Fisa et al. [2] as an example of interventions targeted at reducing RTAs.

The results of this study highlight the significance of establishing safety measures aimed at safeguarding children in their roles as pedestrians, bikers on roadways, or passengers.

Based on the four cases we presented, we would like to make the following recommendations as preventive measures to reduce the occurrence of road fatalities in children:Enforcement of child retention systems, such as safety seats for infants and younger children and booster seats for older children. However, other risk factors need to be taken into consideration since these measures are not 100% efficient.Public awareness/education campaigns for recognizing the risk of alcohol consumption among pedestrians.Appropriate supervision of young children by their parents near traffic, regarding children as pedestrians and bike users.Enforcement of helmet use by motorcyclists.

In addition to these four recommendations, based on all 23 cases we analyzed, we would like to accentuate that employing safe driving conduct, not driving too fast, respecting the rules, not being distracted while driving by eating or talking on the phone, and not being impaired by substance use, can represent the most important rules to follow in order to reduce road fatalities in children.

## 5. Conclusions

Our findings emphasize the tragedy of road fatalities in children and the need to determine the risk factors and prevention strategies to reduce the enormous global crisis involving these vulnerable victims. To successfully reduce road fatalities in children, effective preventive measures need to be considered, with the involvement of political officials, financial funding, and public engagement.

The findings of this study illustrate the importance of studying road deaths in children to identify the risk factors that lead to these fatalities to be able to develop new strategies or to improve the existing ones for preventing RTAs. It is of the utmost importance that all road users have safe conduct, with the emphasis on the driver’s important role. Ensuring safe driving habits, like not speeding or having any distractions or impairments and respecting the rules of the road, needs to represent the focus of campaigns targeted at educating the public. The use of protective measures such as helmets for all motorcycle users, safety belts for any passenger (not only drivers), and child retention systems should represent another important focus. The role of alcohol among pedestrians should not be neglected, and “drunk walking” should also be addressed by preventive measures. Of utmost importance is that parents should supervise children’s activities near the road and that all children should be accompanied.

These recommendations and the public awareness campaigns, along with the help of legislation and enforcement by the authorities, aim at the reduction and elimination of road fatalities by protecting children and other vulnerable road users.

## Figures and Tables

**Figure 1 children-11-01065-f001:**
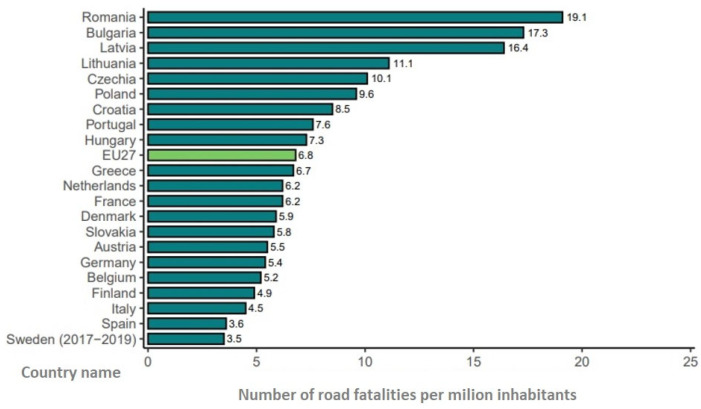
The number of road fatalities in children aged between 0 and 14 per million inhabitants per country in the EU for 2018–2020: light green represents the EU mean (EU27). Source: European Commission (2022) [8].

**Figure 2 children-11-01065-f002:**
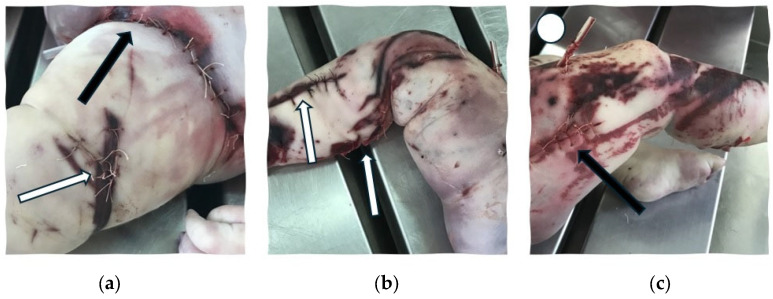
External examination of the body during the medico-legal autopsy: (**a**) Left thigh with excoriations and sutured laceration (white arrow); left inguinal region with sutured laceration and drain tube (black arrow); (**b**,**c**) Right lower extremity with excoriations, sutured lacerations (white arrows), postoperative sutured wound (black arrow) and drain tube (white circle).

**Figure 3 children-11-01065-f003:**
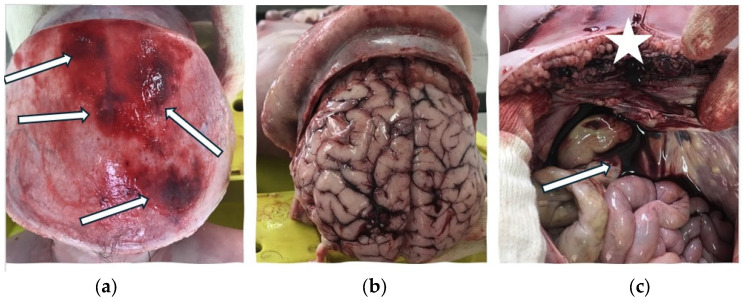
Internal examination of the body during the medico-legal autopsy: (**a**) Hemorrhage in the pericranium (white arrows); (**b**) Cerebral edema; (**c**) Hemoperitoneum (white arrow) and hemorrhage in the pelvic tissues (white star).

**Figure 4 children-11-01065-f004:**
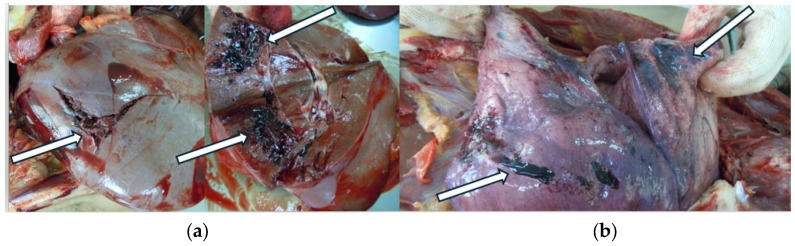
Internal examination of the body during the medico-legal autopsy: (**a**) Hepatic contusions and lacerations (white arrows); (**b**) Pulmonary contusions (white arrows).

**Figure 5 children-11-01065-f005:**
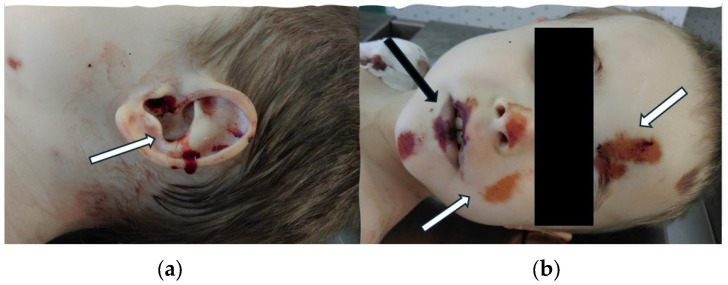
External examination of the body during the medico-legal autopsy: (**a**) Left eat bleeding (white arrow); (**b**) Bruises (black arrow) and excoriations (white arrows) located on the victim’s face.

**Figure 6 children-11-01065-f006:**
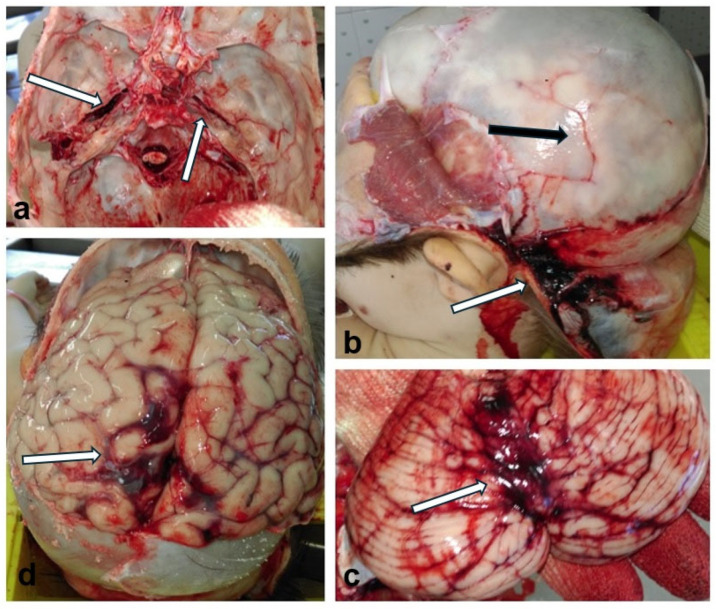
Internal examination of the body during the medico-legal autopsy: (**a**) Fractured skull base (white arrows); (**b**) Fractured skull cap (black arrow), hemorrhage in the pericranium (white arrow); (**c**) Cerebellar hemorrhage (white arrow); (**d**) Subarachnoid hemorrhage (white arrow), cerebral edema.

**Figure 7 children-11-01065-f007:**
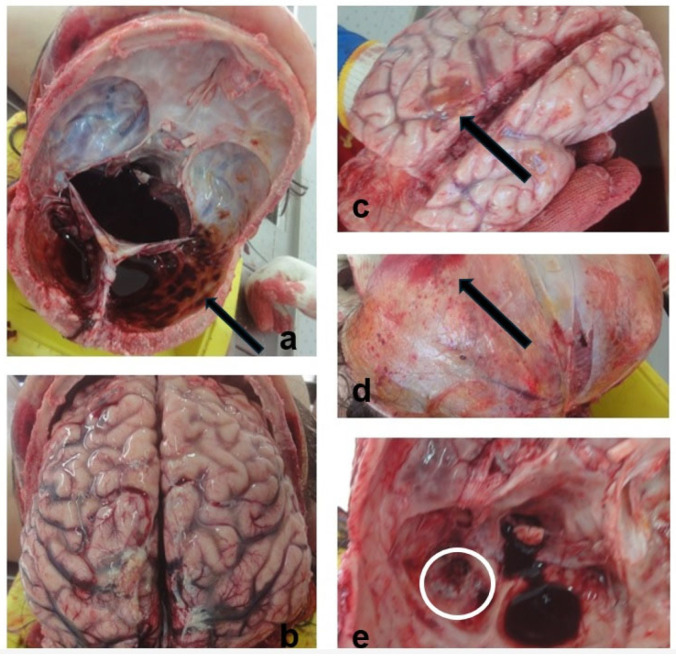
Internal examination of the body during the medico-legal autopsy: (**a**) Skull base with remains of the subdural hematoma (black arrow); (**b**) Cerebral edema; (**c**) Subarachnoid hemorrhage (black arrow); (**d**) Hemorrhage in the pericranium (black arrow); (**e**) Craniectomy orifice in the skull base (white circle).

**Figure 8 children-11-01065-f008:**
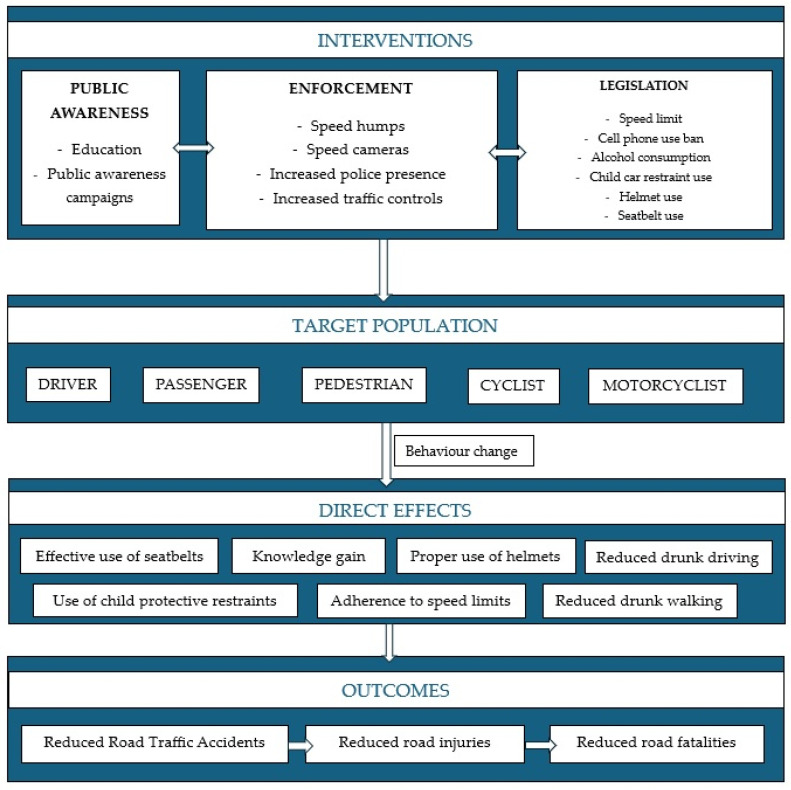
An example of interventions targeted at reducing RTAs by combining public awareness with enforcement and legislation, targeted at all road users. Our adaptation of the model presented by Fisa et al. [2].

**Table 1 children-11-01065-t001:** Annual distribution of people that were fatally injured in RTAs and had a medico-legal autopsy at Timisoara Institute of Legal Medicine (TILM) in Timis county, Romania, in a 5-year period (2017–2021), stratified by age group.

	Age Groups (Years)				
Year	<18	18–50	51–70	>70	Total
2017	9	52	25	11	97
2018	6	57	35	8	106
2019	2	31	27	13	73
2020	3	35	12	9	59
2021	3	27	19	10	60
Total	23 (5.8%)	202 (51.1%)	118 (29.9%)	51 (12.9%)	394 ^1^

^1^ one case represented the autopsy of a fetus.

**Table 2 children-11-01065-t002:** General representation of road fatalities in children that were medico-legal autopsied at TILM in Timis county, Romania, in a 5-year period (2017–2021).

Case Number	Age	Gender	Type of Road User	Circumstances of RTA	Hospitalization	External Injuries	Internal Injuries	Cause of Death	BAC
CASE 1	16	female	front seat passenger	The RTA happened at 1:30 a.m. Collision between two cars (the victim’s car steered into oncoming traffic) Dead at the crash scene	NO	lacerations located on the head, bruises and excoriations located on the upper and lower limbs	hemorrhage in the pericranium, skull fractures, subarachnoid hemorrhage, contusions and lacerations of the brain stem; cervical spine fracture; sternal and rib fractures, contusions and lacerations of the lungs, bilateral hemothorax, laceration of the heart muscle and pericardium, tears of thoracic blood vessels; hemoperitoneum, hepatic lacerations, splenic lacerations, renal lacerations; clavicle fracture, fractures of the os coxae	polytrauma	zero
CASE 2 *	4	male	cyclist	The RTA happened at 7:45 a.m. Collision between a car and the victim (the car entered the street while veering to the right when it hit the victim, who was crossing the street at a crosswalk)	Died in the emergency room	bruises and excoriations located on the face, neck, upper and lower limbs, ear bleeding	hemorrhage in the pericranium, skull fractures, subarachnoid hemorrhage, cerebral, cerebellar and intraventricular hemorrhage	cranio-cerebral trauma	was not required
CASE 3	3	female	pedestrian	The RTA happened at 10:38 a.m. Collision between a car and the victim (the car, while parking, hit the victim, who was alone and unsupervised on the road)	Died in the emergency room	excoriations located on the head and upper and lower limbs, laceration on the thoracic wall, ear bleeding	hemorrhage in the pericranium, skull fractures, subarachnoid hemorrhage, intraventricular hemorrhage; hemothorax, pulmonary lacerations; renal lacerations	polytrauma	zero
CASE 4	17	male	back seat passenger	The RTA happened on a rural road (no other information is available)	YES; 40 days	recent scars on the head and upper limbs	hemorrhage in the pericranium, skull fractures, subdural and extradural hematoma, subarachnoid hemorrhage, cerebral contusions, brain stem contusions; lung contusions; left radius and ulna fractures	polytrauma	NO
CASE 5	17	male	motorcyclist	The RTA happened on an urban road (no other information is available)	YES; 5 days	excoriations and lacerations on the head, bruises and excoriations on upper limbs, bruises on lower limbs	hemorrhage in the pericranium, subdural hematoma, subarachnoid hemorrhage, cerebral contusions, cerebellar and brain stem contusions, intraventricular hemorrhage	cranio-cerebral trauma	NO
CASE 6	1	female	pedestrian	The RTA happened at 7:45 p.m. on a rural road Collision between a car and the victim (the car hit the victim, who was on the grass near the road; the driver lost control of the vehicle) Dead at the crash scene	NO	bruises and excoriations on the head, excoriations on the upper and lower limbs, bruises and excoriations on the thoracic and abdominal wall	hemorrhage in the pericranium; sternal and rib fractures; hemothorax, lung lacerations; hemoperitoneum, hepatic contusions and lacerations, renal lacerations; left femur fracture	polytrauma	was not required
CASE 7 *	3 months	male	back seat passenger	The RTA happened at 2:00 p.m. Collision between a car, a light pole, and a building (the victim’s car was driven by the father of the child who, while overtaking a car, hit a light pole and then was projected into a building) Strapped in an infant car seat	YES; 1 day	bruises and excoriations located on the head, thorax, abdominal wall, and lower limbs; sutured lacerations on lower limbs and external genital organs	hemorrhage in the pericranium; hemoperitoneum; hemorrhage in the pelvic tissues; right femur open fracture, severed femoral artery, left femur open fracture, right tibial open fracture	polytrauma	NO
CASE 8	8	female	pedestrian	The RTA happened at 5:30 p.m. on a rural road Collision between a bus and the victim (the victim was on the road and was hit by a bus)	YES; 26 days	bruises, excoriations and lacerations located on the head, ear bleeding, recent scars, excoriations on the lower limbs	hemorrhage in the pericranium, skull fractures, subarachnoid hemorrhage, cerebral contusions, brain stem contusions	cranio-cerebral trauma	NO
CASE 9	3	male	passenger	The RTA happened at 2:35 p.m. Frontal collision between two cars Dead at the crash scene	NO	bruises and lacerations located on the head, excoriations on the thorax and abdominal wall, bruises and excoriations on upper and lower limbs	hemorrhage in the pericranium, skull fractures, subarachnoid hemorrhage, contusions and lacerations of the brain stem; cervical spine fracture; sternal and rib fractures, contusions and lacerations of the lungs, bilateral hemothorax, hemopericardium, laceration of the heart muscle, tears of thoracic blood vessels; hemoperitoneum, hepatic lacerations, splenic lacerations; clavicle fracture, humerus fracture, femur fracture, tibia and fibula fractures	polytrauma	was not required
CASE 10	6	female	passenger	The RTA happened at 6:49 p.m. Frontal collision between two cars (the other car steered into oncoming traffic and hit the victim’s car) Dead at the crash scene	NO	excoriations and lacerations located on the head, bruises on the upper limbs	hemorrhage in the pericranium, skull fractures, subarachnoid hemorrhage, contusions of the cerebellum and the brain stem, intraventricular hemorrhage;	cranio-cerebral trauma	was not required
CASE 11	4	Female	pedestrian	The RTA happened at 3:20 p.m. on a rural road The victim was hit by a car while crossing the street	YES; 23 days	lacerations and excoriations located on the head, excoriations on the lower limbs	hemorrhage in the pericranium, skull fractures, subarachnoid hemorrhage, cerebral contusions and lacerations, contusions of the brain stem; hepatic lacerations, renal lacerations, pancreatic hematoma	polytrauma	NO
CASE 12 *	17	male	pedestrian	Found dead on the side of a county road at 4:40 a.m. The victim was hit by a bus The bus driver tried to avoid what had seemed to be the body of a dead animal on the right side of the road, but by doing this, he was not able to notice the victim on the left side of the road and could not stop the bus in time to avoid the accident	NO	cranio-dural-cerebral lacerations, excoriations and lacerations on the thoracic anterior wall, abdominal anterior wall, lumbar regions, sacral region, and upper and lower limbs	hemorrhage in the pericranium, numerous skull fractures with bone fragments, cerebellar lacerations; lung contusions; hemoperitoneum, hepatic lacerations and contusions and hemorrhage surrounding the right kidney	polytrauma	1.05 g‰
CASE 13	17	male	pedestrian	The RTA happened at 05:20 a.m. on a rural road The victim was found on the side of the road	YES; 7 days	cranio-dural-cerebral laceration, bruises located on the head, excoriations on the thorax, excoriations, bruises and lacerations on the lower and upper limbs	hemorrhage in the pericranium, skull fractures, subarachnoid hemorrhage, cerebral contusions and lacerations, contusions of the brain stem, intraventricular hemorrhage; cervical spine fracture; rib fractures; colon and mesocolon contusions; clavicle fracture	polytrauma	NO
CASE 14	3	female	pedestrian	The RTA happened at 11:40 a.m. on an urban road Collision between a car and the victim (the car hit the victim who was on the side of the road; the driver lost control of the vehicle)	Died in the emergency room	excoriations on the head, neck and limbs, bruises on the head and lower limbs, ear bleeding, cranio-dural lacerations	hemorrhage in the pericranium, skull fractures, subarachnoid hemorrhage, cerebral contusions and lacerations, contusions of the cerebellum, intraventricular hemorrhage; atlanto-occipital dislocation; bilateral hemothorax; hemoperitoneum; hepatic lacerations, hemorrhage surrounding the right kidney; right femur fracture, open fracture of the right humerus	polytrauma	Was not required
CASE 15	9	female	back seat passenger	The RTA happened at 9:18 p.m. on an urban road (no other information is available)	Died in the emergency room	lacerations and excoriations located on the head, excoriations and bruises on the thorax, abdominal wall, and upper and lower limbs	cervical spine fracture; hemothorax, lung lacerations; hemoperitoneum, hepatic lacerations, pancreatic contusions, hemorrhage surrounding the right kidney, renal lacerations	polytrauma	zero
CASE 16	5	male	back seat passenger	The RTA happened at 04:35 a.m. on a highway Frontal collision between two cars (a car was driving on the wrong side of the highway when it hit the victim’s car; the guilty driver left the crash scene; he was then apprehended by the police and, when tested, he presented with a BAC of 0.39 g‰; three people died in the RTA)	Died on transfer from one hospital to another	Bruises and excoriations located on the head, thorax, and upper and lower limbs	hemorrhage in the pericranium, skull fractures, subarachnoid hemorrhage, subdural hematoma, cerebral contusions and lacerations, intraventricular hemorrhage; atlanto-occipital dislocation; rib fracture, lung lacerations; hemoperitoneum, mesenteric tears, splenic lacerations	polytrauma	was not required
CASE 17 *	17	female	motorcyclist (passenger)	The RTA happened at 9:30 p.m. on an urban road (no other information is available)	YES; 36 days	recent scars on the head, thoracic wall, and upper and lower limbs	hemorrhage in the pericranium, skull fractures, subarachnoid hemorrhage, cerebral, cerebellar and intraventricular hemorrhage	polytrauma	NO
CASE 18	15	male	back seat passenger	The RTA happened at 11:30 p.m. on an urban road (the driver lost control of the vehicle while approaching a bend and, when trying to gain control of the car, it hit a bridge head and then a street lamp) Dead at the crash scene	NO	bruises located on the head, excoriations on the head and thorax	hemorrhage in the pericranium, skull fractures, subarachnoid hemorrhage	cranio-cerebral trauma	zero
CASE 19	14	male	front seat passenger	The RTA happened at 11:30 p.m. on an urban road (the driver lost control of the vehicle while approaching a bend and, when trying to gain control of the car, it hit a bridge head and then a street lamp) Dead at the crash scene	NO	lacerations, bruises, and excoriations located on the head, bruises and excoriations on the upper and lower limbs	cervical spine fracture; rib fractures, lung contusions; hemoperitoneum, hepatic lacerations	polytrauma	zero
CASE 20	14	female	passenger	The RTA happened at 12:40 p.m. on an urban road (no other information is available)	YES; 15 days	bruises located on the head, excoriations on the upper limbs	hemorrhage in the pericranium, skull fractures, subarachnoid hemorrhage, subdural hematoma, intracerebral hematoma, cerebral lacerations	cranio-cerebral trauma	NO
CASE 21	5	male	passenger	The RTA happened on an urban road (no other information is available)	YES; 74 days	cranio-dural-cerebral lacerations	skull fractures, subarachnoid hemorrhage, subdural hematoma, intraventricular hemorrhage, cerebral lacerations	cranio-cerebral trauma	NO
CASE 22	17	male	passenger	The RTA happened at 5:35 p.m. on an urban road (no other information is available) Dead at the crash scene	NO	lacerations, excoriations, and bruises located on the head, bruises and excoriations on the upper and lower limbs	hemorrhage in the pericranium, skull fractures, subarachnoid hemorrhage, intraventricular hemorrhage, cerebral contusions, hemothorax, lung contusions; right radius and ulna fractures	cranio-cerebral trauma	zero
CASE 23	16	male	passenger	The RTA happened at 2:20 a.m. on an urban road (no other information is available) Dead at the crash scene	NO	excoriations located on the head, bruises and excoriations on the upper and lower limbs	hemorrhage in the pericranium, cerebral contusions, intraventricular hemorrhage; bilateral hemothorax, rib fractures, lung contusions; hemorrhage surrounding the spleen and the right kidney; right clavicle fracture	polytrauma	zero

* the four cases are presented in detail below.

## Data Availability

The data presented in this study is available on request from the corresponding author. The data are not publicly available due to confidentiality reasons.

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
