# Peer review of "Road Fatalities in Children Aged 0–17: Epidemiological Data and Forensic Aspects on a Series of Cases in a Single-Centre in Romania"

_children, 2024, doi:10.3390/children11091065_

Round 1

Reviewer 1 Report

Comments and Suggestions for Authors

Dear authors,

I have read your manuscript with pleasure,

I hope that with the following suggestions it can be improved:

in line 48, there is a contradiction with what is said in the abstract;

please insert the citation in lines 64, 75, and 76;

In line 82, where green mobility and related safety devices are discussed, please integrate these aspects with the legislative and traumatological elements; please refer to: https://doi.org/10.3390/su14106160

The introduction is interesting up to line 104 but sometimes repetitive. Please review the entire introduction, maintaining a deductive approach and addressing the topics with greater logical rigor.

Finally, the transition to autopsies is a bit disconnected. Please adequately motivate the transition to this topic, the gap in the study, and the study's aims.

In line 151, please specify the number of autopsies of interest for the study and the percentage compared to autopsies in general and for road accidents.

In paragraph 3.3, there is no indication of using electric scooters. Were they included in another category?

Please clearly indicate the reasons behind the choice to present the 4 cases.

Lines 350-355 are not necessary.

The cases are discussed from 362 to 434, but it is unclear why the authors have included them and why they do not discuss them.

The rest of the discussion is general and not very adherent to the selected sample but to the single case, losing the scientific significance of the article.

What is the reasoning behind the flowchart construction illustrated in Figure 10?

The conclusions are very generic and add nothing new to the scientific literature.

Kind regards

Author Response

Dear Reviewer 1, 

Thank you very much for taking the time to read and review our article. We are very pleased to hear you read our manuscript with pleasure and we hope that the changes we made to the manuscript according to your comments and the other reviewers' comments, have improved the manuscript.

Please find the detailed responses below and the corresponding revisions and corrections are highlighted in the re-submitted file.

Comment 1: in line 48, there is a contradiction with what is said in the abstract;

Response 1: Thank you for pointing this out. However, we want to make some clarifications to what you addressed. In line 48 is noted that Road Traffic Accidents (RTAs) represent the eight leading cause of death for children aged 4 and under. In the abstract, is mentioned that RTAs are the leading cause of premature death in young people aged 5-29 (lines 24-25). Therefore, RTAs represent the first cause of death for people aged 5-29 years old and they also represent the 8th cause of death in children aged 0-5 years old.

Comment 2: please insert the citation in lines 64, 75, and 76;

Response 2: Thank you for addressing this.

The citation for line 64 (now line 72) is the same as for line 67 (now line 75), citation [8]. Lines 62-67 (now 70-75) all have the citation [8]. We considered there was no need to include the citation for every sentence, so we only added the citation in line 67 (now line 75) for lines 62-67 (now line 70-75). We hope this agrees with you.  We did not make any modifications in the manuscript regarding the citation for line 64 (now line 72). If we still need to add the citation, we will gladly do it.

The citations for lines 75 (now line 56) and 76 (now line 57) is citation [5]. We rearranged the ideas in the introduction and some line do not correspond with the old version of the manuscript. We added the citation to the mentioned line. Thank you.

Comment 3: In line 82, where green mobility and related safety devices are discussed, please integrate these aspects with the legislative and traumatological elements; please refer to: https://doi.org/10.3390/su14106160

Response 3: Thank you very much for your suggestion. We read the suggested article with pleasure. We wrote about the location of traumatic injuries without the protection of helmets, in these particular cases (see lines 82-85 in the re-submitted version).

Comment 4: The introduction is interesting up to line 104 but sometimes repetitive. Please review the entire introduction, maintaining a deductive approach and addressing the topics with greater logical rigor.

Response 4: Thank you for your kind words and we are glad you found the introduction interesting. We addressed your comments and we made modifications to the introduction, we rearranged the ideas and we added new paragraphs (lines 108-115), which we highlighted in the manuscript. I hope you will find that what we modified is improving the manuscript's introduction and is satisfactory and reaches your requirements.

Comment 5: Finally, the transition to autopsies is a bit disconnected. Please adequately motivate the transition to this topic, the gap in the study, and the study's aims.

Response 5: We agree with you. There is a gap in the information before the aim of the study and the aim of the study. We added some clarifications to the matter explaining why we studied medico-legal autopsies and why by studying them we can address risk factors for children fatalities in RTAs and therefore find preventive strategies. These modifications are highlighted in the text in lines 108-115. Thank you for this observation. We have also made modifications to the introduction which are highlighted in the manuscript. We hope these bring improvements and reach your requirements. 

Comment 6: In line 151, please specify the number of autopsies of interest for the study and the percentage compared to autopsies in general and for road accidents.

Response 6: Thank you for your suggestion. In the 5-year period we studied (2017-2021) there was a total of 3752 medico-legal autopsies. The number of autopsies for victims of road traffic accidents of any age was 395 (10.5%). The number of road deaths in children was 23 cases (5.8% from all RTAs and 0.61% from all medico-legal autopsies). We made modifications to lines 149-151 and also added this information in the manuscript, in highlight (now lines 159-163): "Between January 1, 2017, and December 31, 2021, a total of 23 medico-legal autopsies involving children aged 17 and under that died in RTAs, were performed. This represents 0.61% of all medico-legal autopsies at TILM in the 5-year period (n=3752). They also re- present 5.8% of all RTA autopsies (n=395)."

Comment 7: In paragraph 3.3, there is no indication of using electric scooters. Were they included in another category?

Response 7: Thank you for your observation. The information is correct. We did not include electric scooters because we did not have this type of road user in our sample: 13 passengers, 7 pedestrians, 1 cyclist and 2 motorcyclists. No electric scooters were seen.

Comment 8: Please clearly indicate the reasons behind the choice to present the 4 cases.

Response 8: Thank you for your suggestion. It is indeed not understandable our reason for illustrating the 4 cases we presented. Out of all 23 cases, these 4 cases represent one type of road user that we saw while analyzing the cases, regarding the risk factors that led to road fatalities. We made modifications to the manuscript adding new information and clarifications regarding all 23 cases. This information can be seen in lines 173-181 and Table 2 which illustrate a general representation of road fatalities in children . We also clarified the reason in choosing the 4 cases in lines 233-247. Thank you very much for suggesting this and we hope the new information we provided has improved the manuscript.

Comment 9: Lines 350-355 are not necessary.

Response 9: Thank you for this suggestion. As stated by you, after our analysis we also agreed with you that these lines are not necessary for the manuscript, so we deleted them. We made modifications in highlight to the citations in the manuscript. Thank you for your recommendation that improves our manuscript.

Comment 10: The cases are discussed from 362 to 434, but it is unclear why the authors have included them and why they do not discuss them.

Response 10: Thank you for our observation. We made modifications in the manuscript that we highlighted and we explained why we chose these 4 cases to present and what makes them distinguishable from the others (lines 233-247). We also added new information regarding all 23 cases that we hope you will consider that it adds value to our manuscript. New modifications are seen in lines 173-181 and Table 2. In the Discussion chapter we added new information regarding all 23 cases (lines 401-424). I hope this improves this chapter and adds value to the manuscript.

Comment 11: The rest of the discussion is general and not very adherent to the selected sample but to the single case, losing the scientific significance of the article.

Response 11: Thank you for your comment. We added new information to this chapter in lines 394-397 and 401-424 that we hope improves the manuscript. However we consider that the rest of the discussion, after the discussion of the 4 particular cases, offers a background and also an opening towards information about preventive measures and specific interventions in order to reduce road deaths. We also added new information in lines 504-507 and lines 532-536. There is a need to address vulnerable road users but also the general population and all types of road users with interventions targeted at reducing RTAs. We believe that the recommendations we made derive directly from the cases we discussed (this also being the reason we discussed these cases), and represent important preventive measures that can reduce road fatalities in children. Moreover, in the new lines we made a new recommendation derived from all 23 cases. According to your comment we made  modifications to this part of the discussion and we hope you will find them bringing more logic to the discussion part.

Comment 12: What is the reasoning behind the flowchart construction illustrated in Figure 10?

Response 12:  Thank you for this question. We believe that the flowchart illustrated by Figure 10 offers important information regarding specific interventions targeted at reducing RTAs. These interventions are addressed at the general population and are specific to types of road user. Although is it adapted from Fisa et. al (citation [2]), it is our modified version that addresses the importance of alcohol consumption among pedestrians (even as children) and leads to a direct effect such as reduced drunk walking. This alone makes our model of flowchart important because it brings into attention alcohol consumption in pedestrians. Another reason we would like the flowchart to stay in our article is the fact that all interventions needs to be aimed at all types of road users, none of them being too specific to one type of road user, for example seatbelt use needs to be taken into account by all passengers of a car not only by the driver or the front-seat passenger. Moreover, all the interventions need to be taken into account together having a synergetic effect. Only by the combined effect of interventions addressed at the targeted population, effects can be obtained, hopefully with reducing road deaths.

Comment 13: The conclusions are very generic and add nothing new to the scientific literature.

Response 13: Thank you for your observations and we regret you feel this way. However, we hope we succeeded to improve the conclusions with the new lines we introduced by pointing the specific risk factors we identified in our research and the deduced preventive measures. We highlighted the modifications (lines 543-556).

Thank you very much for your comments and observations.

Your review was of high value  to us and we deeply appreciate your feedback.

We hope the modifications we made reach your requirements and bring improvements to the manuscript.

King regards, 

The authors

Reviewer 2 Report

Comments and Suggestions for Authors

Thank you for the opportunity to review for this manuscript entitled: Road Fatalities in Children Aged 0-17: Epidemiological Data 2 and Forensic Aspects On A Series of Cases in A Single-Centre 3 in Romania.

I found the manuscript well written and informative. I have little overall comments, but have some more detailed ones as per below:

Page 1:
The numbers for the authors affiliations, seems to have a smaller ‘1’ compared to the other numbers. I appreciate this may be an editorial issue. Please also note that some of the numbers have a full stop after them, while others do not.

Line 31: I would suggest that ‘involving’ should be ‘involved’.

Line 70-71: This sentence is too strongly worded: “Road traffic deaths will become the seventh main cause of death worldwide by 2030 70 if urgent action is not taken” it could be reformulated to: “Road traffic deaths has been predicted to become the seventh main cause of death worldwide by 2030 70 if urgent action is not taken”

Line 71-72: May I suggest to re-write this sentence: “The main risk factors in the happening of RTAs are considered the following: having an urban speed limit higher than 50 km/h…..” to: “The main risk factors of RTAs are considered as follows: having an urban speed limit higher than 50 km/h, a…..”

Line 109-110: May I suggest this sentence: “to all road deaths, disregarding the period of time from the accident until the victims’ death.”  is re-written to: “to all road deaths, disregarding the period of survival from the accident until the victims’ death.”  

Line 125-126: May I suggest this sentence: “RTAs victims are accounted for violent deaths; therefore, a medico-legal autopsy is performed” is re-written to: “RTAs victims are accounted for as violent deaths; therefore, a medico-legal autopsy is performed”

Line 134-135: “(age of the victims was not between 0 and 134 17 years old)” could be “n (younger than 18 years old) – as per Table 1.

Line 146: “were collected and introduced into an Excel sheet and analyzed using simple mathematical tools” Can this be changed to: “were collected and entered into an Excel spreadsheet and analyzed using descriptive statistics”

Line 166: for this sentence: “Preschoolers accounted for 10 cases distributed equally” be changed to “Preschoolers (under the age of 7) accounted for 10 cases distributed equally”. Reason is that what is considered preschool age is different between countries, and apparently for Romania this would be 7 years?

Line 178: “Between road fatalities, a cyclist was noted” please change to: “Between road fatalities, one cyclist was noted”

Line 204: I would suggest this section is started slightly differently, such as “ Case presentations, have been selected to present four suggestive…..”

Line 221: There is a space too much before “Additionally”

Line 219: It is understood but not well stated, what is meant by “due to altered general state”. This sentence could be rewritten more simply to: “. On admission, an emergency orotracheal intubation with mechanical ventilation was performed.”

Line 228-230: Could this sentence be re-written to: “However, the postoperative recovery was unfavourable and the following morning the patient presented with cardiac arrest that did not respond to resuscitation maneuvers.

Line 242: I am not a pathologist, so I am unsure if the word “excoriations” is correct, can this please be checked by the authors. This word is also used elsewhere in the paper.

Line 255-256: I think this sentence would be better rewritten as: “From the police investigation. It was detailed that he was hit by a bus in a RTA and dies at the crash scene”.

Line 258-259. This sentence: “A medico-legal autopsy was mandatory to establish the cause of death of the victim” is not needed as this has been explained several times before. I would instead suggest to start it with: “ As part of the police investigation, the police asked for the determination of BAC”.

Line 259-260: I would like to think a medico-legal autopsy will always be ‘complete and thorough’ hence these words are unnecessary. I would suggest instead: “The following day, a medico-legal autopsy was performed at TILM”

Line 263: It is unsure what is meant by “at this level” – can this be clarified?

Line 266-267: “a closed fracture of 266 the left tibia were also discovered” would sound better as: “a closed fracture of 266 the left tibia were also determined”

Line 289-290: “the child was critical having cerebrospinal fluid leaks through the ears and nose” could be rewritten to: “the child was in a critical condition having cerebrospinal fluid leakage through the ears and nose”

Line 350: I do not think this sentence fits into the voice of this article: “RTAs represent the price humanity must pay for modernization [21].” And could be completed deleted without any loss.

Line 382: “Just like for drivers, a pedestrian’s risk of crash involvement….” I would suggest: Equally for drivers, a pedestrian’s risk of crash involvement….”

Line 390: change this to: “The third case we presented, a four-year-old cyclist hit by a car”

Line 395-397: This sentence: “Distraction of the driver, including talking on a cell phone, texting or performing other tasks such as eating while driving was associated with road fatalities in children.” I am not sure where is coming from – is this a lesson learned from this study or is there missing a reference?

Line 403-405: “Bicycle helmet is not mandatory because of the possibility that it will strongly decrease bicycling, however, it may be needed for protection [27]. – unless checking the reference, it is not clear this is referencing a study from Netherlands, so is the same the case for Romania? 

Reference number 26 seems to have been mixed up.

Comments on the Quality of English Language

Please see above. 

Author Response

Dear Reviewer 2, 

Thank you very much for your kind words. We also appreciate we had the opportunity for you to review our manuscript. We hope we made the appropriate modifications to the manuscript, as suggested by you and the other reviewers, and that this modifications were pertinent and improved the quality of the manuscript. Moreover, we hope they reached your expectations and addressed your comments.

Please find the detailed responses below. The corresponding revisions and corrections were highlighted in the re-submitted file.

Comment 1: Page 1: The numbers for the authors affiliations, seems to have a smaller ‘1’ compared to the other numbers. I appreciate this may be an editorial issue. Please also note that some of the numbers have a full stop after them, while others do not.

Response 1: Thank you very much for pointing this out. It was our mistake and we fully acknowledge it. We are very sorry it happened. We modified in the manuscript using all the numbers for affiliations in superscript. We highlighted it in the manuscript. Moreover, we are sorry about the full stop that was after numbers 6 and 7. We deleted them in the manuscript. This was also highlighted in the text. We thank you for your observation and we highly admire your keen eye in spotting the mistakes.

Comment 2: Line 31: I would suggest that ‘involving’ should be ‘involved’.

Response 2: Thank you very much for your comment. We agree with your proposal in replacing "involving" with "involved". Additionally, we also made another modification so the phrase that is written in lines 30-32 is now: "Of all medico-legal autopsies in the 5-year period, 23 cases (5.8%) involved road fatalities in children aged 17 and under. " We hope you agree with our modification and that we improved the above mentioned lines. We made the modifications in highlight in the manuscript.

Comment 3: Line 70-71: This sentence is too strongly worded: “Road traffic deaths will become the seventh main cause of death worldwide by 2030 70 if urgent action is not taken” it could be reformulated to: “Road traffic deaths has been predicted to become the seventh main cause of death worldwide by 2030 70 if urgent action is not taken”

Response 3: Thank you very much for your suggestions. It is indeed much appreciated since it softens the whole phrase and makes it sound better. We made the suggested modification and we highlights it in the manuscript. Thank you. Lines 70-71 (now line 78-79 in the re-submitted manuscript): "Road traffic deaths had been predicted to become the seventh main cause of death worldwide by 2030 if urgent action is not taken. "

Comment 4: Line 71-72: May I suggest to re-write this sentence: “The main risk factors in the happening of RTAs are considered the following: having an urban speed limit higher than 50 km/h…..” to: “The main risk factors of RTAs are considered as follows: having an urban speed limit higher than 50 km/h, a…..”

Response 4: Thank you for your comment. We re-wrote the sentence as you suggested. This is highlighted in the manuscript. Lines 71-72 (now lines 52-53): "The main risk factors of RTAs are considered as follows:...."

Comment 5: Line 109-110: May I suggest this sentence: “to all road deaths, disregarding the period of time from the accident until the victims’ death.”  is re-written to: “to all road deaths, disregarding the period of survival from the accident until the victims’ death.”  

Response 5: Thank you for your suggestion. We agree. We modified in the manuscript as you suggested and we highlighted the modifications. Lines 109-110 (now lines 120-121): "to all road deaths, disregarding the period of survival from the accident until the victims’ death."

Comment 6: Line 125-126: May I suggest this sentence: “RTAs victims are accounted for violent deaths; therefore, a medico-legal autopsy is performed” is re-written to: “RTAs victims are accounted for as violent deaths; therefore, a medico-legal autopsy is performed”

Response 6: Thank you for your suggestion. We re-wrote the sentence as you suggested and we highlighted the modifications. Lines 125-126 (now lines 137-138): "RTAs victims are accounted for as violent deaths; therefore, a medico-legal autopsy is performed"

Comment 7: Line 134-135: “(age of the victims was not between 0 and 134 17 years old)” could be “n (younger than 18 years old) – as per Table 1.

Response 7: Thank you for your suggestion. We re-wrote the sentence as you suggested and we highlighted the modifications. Lines 134-135 (now line 145):  "...(were not younger than 18 years old)..." 

Comment 8: Line 146: “were collected and introduced into an Excel sheet and analyzed using simple mathematical tools” Can this be changed to: “were collected and entered into an Excel spreadsheet and analyzed using descriptive statistics”

Response 8: Thank you for your suggestion. We re-wrote the sentence as you suggested and we highlighted the modifications.  Lines 146-147 (now lines 157-158): "...were collected and entered into an Excel spreadsheet and analyzed using descriptive statistics"

Comment 9: Line 166: for this sentence: “Preschoolers accounted for 10 cases distributed equally” be changed to “Preschoolers (under the age of 7) accounted for 10 cases distributed equally”. Reason is that what is considered preschool age is different between countries, and apparently for Romania this would be 7 years?

Response 9: Thank you for your observations. Indeed in Romania, preschool age is under the age of 7.  We agree with the modifications and we highlighted them in the manuscript. Line 167-168 (now line 187): "Preschoolers (under the age of 7) accounted for 10 cases distributed equally in the age groups 0-3 and 4-6 years."

Comment 10: Line 178: “Between road fatalities, a cyclist was noted” please change to: “Between road fatalities, one cyclist was noted”

Response 10: Thank you for your comment. We made the modification and we highlight it in the text. Line 180 (now line 197): "Between road fatalities, one cyclist was noted."

Comment 11: Line 204: I would suggest this section is started slightly differently, such as “ Case presentations, have been selected to present four suggestive…..”

Response 11: Thank you for your suggestions. We made modifications to the mentioned line that we highlighted in the manuscript. Line 206 (now line 224): "As case presentations, four suggestive cases have been selected to present, "

Comment 12: Line 221: There is a space too much before “Additionally”

Response 12: Thank you for your suggestion. We deleted the second space. This is now line 231. (You were referring to line 211 not 221). We highlighted the word "Additionally".

Comment 13: Line 219: It is understood but not well stated, what is meant by “due to altered general state”. This sentence could be rewritten more simply to: “. On admission, an emergency orotracheal intubation with mechanical ventilation was performed.”

Response 13: We appreciate your suggestion. We made the modifications and we highlighted the lines (lines 255-256).

Comment 14: Line 228-230: Could this sentence be re-written to: “However, the postoperative recovery was unfavourable and the following morning the patient presented with cardiac arrest that did not respond to resuscitation maneuvers.

Response 14: Thank you for your suggestion. We modified the phrase as you suggested (lines 265-267).

Comment 15: Line 242: I am not a pathologist, so I am unsure if the word “excoriations” is correct, can this please be checked by the authors. This word is also used elsewhere in the paper.

Response 15: Thank you for your inquiry. Yes, the term excoriation is correct, it means a type of lesion on the skin where the skin is abraded following the action of a traumatic agent. Please see https://www.humanitas.net/wiki/first-aid/excoriation-of-the-skin/

Comment 16: Line 255-256: I think this sentence would be better rewritten as: “From the police investigation. It was detailed that he was hit by a bus in a RTA and dies at the crash scene”.

Response 16: Thank you for your suggestion. We re-wrote the lines 292-293 as you suggested and we highlighted them: "From the police investigation, it was detailed that he was hit by a bus in a RTA and died at the crash scene"

Comment 17: Line 258-259. This sentence: “A medico-legal autopsy was mandatory to establish the cause of death of the victim” is not needed as this has been explained several times before. I would instead suggest to start it with: “ As part of the police investigation, the police asked for the determination of BAC”.

Response 17: Thank you for your suggestion. We agree with you. We made the modification and we highlighted the line 298: "As part of the police investigation, the police asked for the determination of BAC. "

Comment 18: Line 259-260: I would like to think a medico-legal autopsy will always be ‘complete and thorough’ hence these words are unnecessary. I would suggest instead: “The following day, a medico-legal autopsy was performed at TILM”

Response 18: Thank you for your observations. Indeed, a medico-legal autopsy is always thorough and complete, so the addition of those two words was unnecessary, as you suggested. We deleted them. Line 299:  "The following day, a medico-legal autopsy was performed at TILM."

Comment 19: Line 263: It is unsure what is meant by “at this level” – can this be clarified?

Response 19: Thank you for your observation. We wanted to say that bone fragments were seen at the site of the severe lacerations on the cephalic extremity. However, we deleted the words "at this level", since we now consider them not necessary. I hope you agree with us. (line 302)

Comment 20: Line 266-267: “a closed fracture of 266 the left tibia were also discovered” would sound better as: “a closed fracture of 266 the left tibia were also determined”

Response 20: Thank you for your suggestion. We replaced the word "discovered" with the word "determined" (line 306).

Comment 21: Line 289-290: “the child was critical having cerebrospinal fluid leaks through the ears and nose” could be rewritten to: “the child was in a critical condition having cerebrospinal fluid leakage through the ears and nose”

Response 21: Thank you for your suggestions. We made the modification as you suggested and highlighted them in the manuscript (lines 330-332: "At the hospital the child was in a critical condition having cerebrospinal fluid leakage through the ears and nose.").

Comment 22: Line 350: I do not think this sentence fits into the voice of this article: “RTAs represent the price humanity must pay for modernization [21].” And could be completed deleted without any loss.

Response 22: Thank you for your honest suggestion. This is in accordance with another reviewer's opinion. Therefore, we deleted lines 350-355.

Comment 23: Line 382: “Just like for drivers, a pedestrian’s risk of crash involvement….” I would suggest: Equally for drivers, a pedestrian’s risk of crash involvement….”

Response 23: Thank you for your comment. We made the modification you suggested, that is highlighted in the line 448: "Equally for drivers, a pedestrian’s risk of crash involvement ..."

Comment 24: Line 390: change this to: “The third case we presented, a four-year-old cyclist hit by a car”

Response 24: Thank you for your suggestions. We made the modification. Line 456: "The third case we presented, a four-year-old cyclist hit by a car, draws attention to ..." (highlighted in the manuscript).

Comment 25: Line 395-397: This sentence: “Distraction of the driver, including talking on a cell phone, texting or performing other tasks such as eating while driving was associated with road fatalities in children.” I am not sure where is coming from – is this a lesson learned from this study or is there missing a reference?

Response 25: Thank you for your comment. With new information added (lines 461-464 and Table 2 and all the lines that are highlighted in the Results and Discussion sections), this phrase makes more sense. The authors wanted to highlight that in addition to parents lack of supervision for children outdoor activities, the driver can also have activities that are causing distraction without much paying attention to the road, which is the case. In addition to the small stature of children and their lack of awareness of road dangers, these are the pathway to a fatal RTA. The reference for lines 395-397 (now lines 464-466) is [11]. This reference is the same for the next 2 lines, so we believed there was no need to include the reference twice. I hope you agree with us.

Comment 26: Line 403-405: “Bicycle helmet is not mandatory because of the possibility that it will strongly decrease bicycling, however, it may be needed for protection [27]. – unless checking the reference, it is not clear this is referencing a study from Netherlands, so is the same the case for Romania? 

Response 26: Thank you for your observation. It is indeed confusing. We deleted that idea and we replaced it with the following "In Romania, bicycle helmet is mandatory for all cyclists with the age under 16 years old. [28]." (lines 472-473) We hope it clarifies the situations and improves the manuscript.

Comment 27: Reference number 26 seems to have been mixed up.

Response 27: Thank you for your comment. The first time we used reference [26] was in line 384. However, in this revised version of the manuscript, we modified the references since we added new ones. We hope you would consider it is now improved.

Thank you very much for reviewing our manuscript.

We hope the modifications we made are agreeing with you. We tried to take into account all the suggestions you had and to make all the necessary modifications.

Kind regards, 

The authors

Reviewer 3 Report

Comments and Suggestions for Authors

The authors reviewed the forensic autopsies of child fatal motor vehicle collision cases performed at Timisoara Institute of Legal Medicine. As the authors mentioned, the assessment of risk factors leading to fatal motor vehicle collision is important to establish effective preventive measures. Therefore, the review of real-world cases, especially forensic autopsy cases, is valuable.

However, the sample size is too small as 23. The obtained risk factors or autopsy results are well accordance with the issues widely understood. Therefore, this study lacks of novelty.  Also, this study lacks of valuable information for readers in worldwide. 

Author Response

Dear Reviewer 3,

Thank you very much for taking the time to read and review our article. We respect your opinion but we feel sorry you feel this way.

As you said, in this manuscript we analyzed medico-legal autopsy records from Timisoara Institute of Legal Medicine in a 5-year period (2017-2020) in order to identify child victims that died in Road Traffic Accidents (RTAs). Seeing that Romania had the highest mortality rate for children's death in 2018-2020, per million inhabitants, per country in the European Region, we felt it was imperatively necessary to address this matter and try to understand why, how and what can be done in order to modify these numbers.

To our knowledge this is the only study that addresses road deaths in children in Romania by analyzing medico-legal autopsies. Only by doing so, we can obtain a valuable insight into the circumstances and the mechanisms of RTAs, seeing that all road-related deaths need to be medico-legal autopsied according to Romanian laws.

In this study, by analyzing the before-mentioned records, we obtained 23 cases of road deaths in children. These cases represent 5.8% of all RTAs records and 0.61% of all medico-legal autopsies. We studied all 23 cases in detail by assessing the age and gender of the victims, the type of road users, the circumstances of the accidents, the hospitalization period, the traumatic injuries (external and internal), the cause of death and the blood alcohol concentration. In the re-submitted manuscript, we presented in Table 2, all 23 cases in detail. Moreover, we selected four cases that stood out and illustrated in the manuscript four case presentations. The reason behind choosing these four cases is explained in the re-submitted manuscript. We chose these cases because each of them represents a different type of road users and the RTAs involve the combinations of more risk factors. By presenting these four cases, we think we illustrate the association between risk factors and try to find preventive measures. In the Discussion chapter, we integrated all the 23 cases with each other in order to find the connecting link and see how these accidents could have been avoided. Additionally, for each of the four cases analyzed, we made a discussion section and integrated them in the existing literature. 

We understand that you think this study lack novelty. However, we think that although 23 is a low number, by offering all the details and by analyzing the circumstances of the accidents, we gained a whole vision as to why these fatal accidents happened. The novelty of the manuscript is in fact the 23 cases that we presented and the fact that we presented in details the internal and external injuries, and we think that for pathologists this manuscript will be of interest. Moreover, we think that not only pathologists will consider this topic hot, but other researches that study road traffic accidents and the circumstances that lead to them too, seeing that we display here real cases that are overanalyzed and tried to understand all the dynamics that led to such fatal accidents. We think this offers valuable information for readers worldwide because Romania is the first at child mortality rates in RTAs and it is very important to understand why.

We think the "why" is due to the reckless, unsafe and aggressive driving that is in Romania, seeing that in our cases we found driving on wrong lanes on highways, driving while impaired by alcohol, driving into incoming traffic, overtaking of cars without safely doing it so (with the driver's child in the car), losing control of the vehicle maybe be over speeding or distraction of the driver and hitting a child or an object (street lamp, bridge head or building). We also saw that lack of supervision of children from the parents have devastating effects and more emphasis needs to be put on this topic. Unsupervised children near the road become targets to impaired/distracted/hurry drivers. Moreover, this study brings into attention alcohol consumption among pedestrians, which is not that discussed, but it is in fact a risk factor that needs to be taken into consideration even in children, seeing that adolescents can seek risky behaviors.

We think the recommendations we made in the Discussion chapter derive from the cases we presented, all 23, and they are very valuable to others because although they are not new information, they make the existing information more potent because they illustrate once again what happens and what can be done in case of a fatal RTA. The most important thing is to make people aware of the dangers driving/waking/biking pose for children, when all the conditions mentioned before, collide. 

Thank you very much for reviewing the manuscript.

We hope that the changes we made to the manuscript have improved the quality and that you will find them suitable and they reached your expectations.

King regards, 

The Authors

Reviewer 4 Report

Comments and Suggestions for Authors

Dear Authors, I have read your article with interest.

In my opinion, the value of your article is determined by a detailed description of the specific injuries sustained in an accident and an analysis of the technology of pathological and anatomical procedures.

I quite understand the choice to describe the specific 4 cases of child deaths. These are very different and, at the same time, very typical cases. The Authors described in detail and professionally the set of injuries that caused the death of the participants in the accident.

My comments and suggestions on the article:

1. I think that the article should focus on a detailed analysis of all deaths involving children. 23 deaths are just a little. You have the opportunity to describe in detail all 23 cases. Let it not be as detailed as in the above 4 cases. For example, you can present information in tabular form. Probably, it would be possible to add available data on specific accident situations to this table (if this information is available to you).

2. The scheme of Figure 10 in your article largely duplicates the scheme given in the article Fisa R, Musukuma M, Sampa M, Musonda P, Young T. Effects of interventions for preventing road traffic crashes: an overview of systematic reviews. BMC Public Health. 2022 Mar 16;22(1):513. The differences are only in form, but not in essence. This scheme is, of course, correct, but it is very simple and does not carry a significant scientific burden. I understand that the purpose of your article is to analyze the epidemiological causes of death of children in road accidents. And I think that this scheme is rather an unnecessary burden, but not a necessity for this study.

3. Perhaps the information in Figure 2 and Figure 3 can be presented in a simple tabular form. These diagrams do not add additional value to the article.

4. Your statistics (lines 162-163 and fig. 2) does not correlate well with the statistics given in the text of the article (lines 25-26, 51 and lines 350-351). 10/23 = 43% - deaths of children under the age of 7 in your case (Timisoara, Romania) and 186300/950000 = 19.5% - deaths of children under the age of 9 (global statistics). It would be great if you could explain this difference.

5. The Conclusions section should still be more informative. 6 lines of text is too little.

In general, the your article, in my opinion, may well be brought to a good quality level worthy of publication in reputable prestigious journals.

Author Response

Dear Reviewer 4,

Thank you very much for reviewing this manuscript and for your kind words in praising our study. We are glad you read this article with interest. and we hope that with these new modifications we managed to improve the quality of this manuscript. We hope it reaches your expectations and that you consider your comments addressed. 

Please find the detailed responses below and the corresponding revisions and corrections are highlighted in the re-submitted file.

Comment 1: I think that the article should focus on a detailed analysis of all deaths involving children. 23 deaths are just a little. You have the opportunity to describe in detail all 23 cases. Let it not be as detailed as in the above 4 cases. For example, you can present information in tabular form. Probably, it would be possible to add available data on specific accident situations to this table (if this information is available to you).

Response 1: Thank you for your comment and suggestions. The aim of this study was to understand and isolate the deaths of children that occur in road traffic accidents, seeing that Romania had the highest mortality rate for children in 2018-2020 in road fatalities per million inhabitants per country in the European Region. We could analyze road deaths in children by assessing medico-legal autopsies since in Romania all traumatic deaths are mandatory to be medico-legal autopsied. In the re-submitted form of the manuscript, we took your suggestion and described in detail all 23 cases as you can see in Table 2 sub-chapter 3.2. Summary and general characteristics of all 23 road fatalities in children aged 0-17 years old (lines 173-182). We also illustrated the circumstances that led to the accidents, with as much details as we had in the autopsy records. We hope this adds new value to the manuscript and addresses your concerns.

Comment 2: The scheme of Figure 10 in your article largely duplicates the scheme given in the article Fisa R, Musukuma M, Sampa M, Musonda P, Young T. Effects of interventions for preventing road traffic crashes: an overview of systematic reviews. BMC Public Health. 2022 Mar 16;22(1):513. The differences are only in form, but not in essence. This scheme is, of course, correct, but it is very simple and does not carry a significant scientific burden. I understand that the purpose of your article is to analyze the epidemiological causes of death of children in road accidents. And I think that this scheme is rather an unnecessary burden, but not a necessity for this study.

Response 2: Thank you for your comment. We understand you view of this scheme but we believe that the flowchart illustrated by Figure 10 (now Figure 8) offers important information regarding specific interventions targeted at reducing RTAs. These interventions are addressed at the general population and are specific to types of road user. Although is it adapted from Fisa et. al (citation [2]), it is our modified version that addresses the importance of alcohol consumption among pedestrians (even as children) and leads to a direct effect such as reduced drunk walking. This alone makes our model of flowchart important because it brings into attention alcohol consumption in pedestrians. Another reason we would like the flowchart to stay in our article is the fact that all interventions needs to be aimed at all types of road users, none of them being too specific to one type of road user, for example seatbelt use needs to be taken into account by all passengers of a car not only by the driver or the front-seat passenger. Moreover, all the interventions need to be taken into account together having a synergetic effect. Only by the combined effect of interventions addressed at the targeted population, effects can be obtained, hopefully with reducing road deaths. We hope this agrees with you.

Comment 3: Perhaps the information in Figure 2 and Figure 3 can be presented in a simple tabular form. These diagrams do not add additional value to the article.

Response 3: Thank you for your suggestion. We agree with you. In the re-submitted form you will see that data from Figure 2 and Figure 3 are illustrated in Table 3. Thank you for your suggestion of improving the manuscript.

Comment 4: Your statistics (lines 162-163 and fig. 2) does not correlate well with the statistics given in the text of the article (lines 25-26, 51 and lines 350-351). 10/23 = 43% - deaths of children under the age of 7 in your case (Timisoara, Romania) and 186300/950000 = 19.5% - deaths of children under the age of 9 (global statistics). It would be great if you could explain this difference.

Response 4: Thank you for your comment. You have a keen eye for details. These statistics does not correlate well with each other because: in our case we illustrate road death in children in 5 years by analyzing medico-legal autopsy records at Timisoara Institute of Legal Medicine (n=23) with 10 children with the age under 7 years old; the international statistics state that  "globally 186 300 children aged 9 years and under die from RTAs each year" and that "950 000 children aged 18 and under die from injuries and violence worldwide" and this includes all kind of violence and injuries not only those related to road traffic accidents. In our study we did not include other types of violent deaths in children (deaths not related to road traffic accidents). I hope this clarifies the situation and addresses your concern.

Comment 5: The Conclusions section should still be more informative. 6 lines of text is too little.

Response 5: Thank you for your comment. We added new information to the Introduction explaining why we chose to study medico-legal autopsy records in order to analyze road traffic accidents in children. We also added new information in the Results section, illustrating all 23 cases of road fatalities in children and the circumstances that led to the accidents. With this new information, we also added new lines in the Discussion chapter, discussing all 23 cases by analyzing the circumstances that led to the accidents and trying to obtain preventive measures targeted at reducing RTAs. Lastly, in the Conclusion chapter, we added new information regarding the conclusions we draw by analyzing all 23 cases, such as safe driving conduct, supervision of children, use of protective measures, drunk walking. These are explained in details in lines 545-557. 

Thank you very much for reviewing this manuscript. 

We hope with the modifications we made, we improved the quality of the manuscript and that this new form will reach your expectations.

We also hope, as you said, that this re-submitted version of the manuscript is getting close to good quality level worthy of publication in reputable prestigious journals.

King regards, 

The Authors

Round 2

Reviewer 1 Report

Comments and Suggestions for Authors

Dear authors,

The paper has been significantly improved and, in my opinion, is now ready for publication.

Best regards.

Reviewer 3 Report

Comments and Suggestions for Authors

The authors reviewed the forensic autopsies of child fatal motor vehicle collision cases performed at Timisoara Institute of Legal Medicine. As mentioned in the first review, the assessment of risk factors leading to fatal motor vehicle collision based on forensic autopsy cases is valuable.

However, because the obtained risk factors or autopsy results are well accordance with the previously known issues, this study lacks of novelty. Furthermore, as the study is based on the observation of small cases of forensic autopsies, the study produced little scientific creative results.